# CONFLICTBANK: Supplementary Information

**Zhaochen Su**[1], **Jun Zhang**[1], **Xiaoye Qu**[2], **Tong Zhu**[1],
**Yanshu Li**[1], **Jiashuo Sun**[2], **Juntao Li**[1], **Min Zhang**[1], **Yu Cheng**[3]
[1]Institute of Computer Science and Technology, Soochow University, China
[2]Shanghai AI Laboratory, [3]The Chinese University of Hong Kong
{suzhaochen0110,junzhang20030309,gasolsun36}@gmail.com
{ljt,minzhang}@suda.edu.cn; xiaoye@hust.edu.cn;
tzhu1997@outlook.com; chengyu@cse.cuhk.edu.hk

# Contents

Submitted to the 38th Conference on Neural Information Processing Systems (NeurIPS 2024) Track on Datasets and Benchmarks. Do not distribute.

# A Discussion

## A.1 Code Access

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

## E  Datasheet

We follow the documentation frameworks provided by Wang et al. [2023b].

### E.1  Distribution

**For what purpose was the dataset created?**

- Please refer to Appendix A.2.

**Who created the dataset (e.g., which team, research group) and on behalf of which entity (e.g., company, institution, organization)?**

- The dataset is jointly developed by a collaborative effort in the author list.

**Composition/collection process/preprocessing/cleaning/labeling and uses:**

- The answers are described in our paper as well as website: `https://github.com/zhaochen0110/conflictbank`.

### E.2  Distribution

**Will the dataset be distributed to third parties outside of the entity (e.g., company, institution, organization) on behalf of which the dataset was created?**

- Yes, the dataset is open to the public.

**How will the dataset will be distributed (e.g., tarball on website, API, GitHub)?**

- The dataset will be distributed through Hugging Face and the code used for developing baseline models through GitHub.

**When will the dataset be distributed?**

- It has been released now.

---

[2]`https://github.com/hiyouga/LLaMA-Factory.`

**Will the dataset be distributed under a copyright or other intellectual property (IP) license, and/or under applicable terms of use (ToU)?**

- Our dataset will be distributed under the CC BY-SA 4.0 license.

### E.3 Maintenance

**How can the owner/curator/manager of the dataset be contacted (e.g., email address)?**

- The owner/curator/manager(s) of the dataset can be contacted through following emails: Zhaochen Su (suzhaochen0110@gmail.com) and Prof. Juntao Li (ljt@suda.edu.cn).

**Is there an erratum?**

- No. If errors are found in the future, we will release errata on the main web page for the dataset (https://github.com/zhaochen0110/conflictbank).

**Will the dataset be updated (e.g., to correct labeling errors, add new instances, delete instances)?**

- Yes, the datasets will be updated whenever necessary to ensure accuracy, and announcements will be made accordingly. These updates will be posted on the main web page for the dataset (https://github.com/zhaochen0110/conflictbank).

**If the dataset relates to people, are there applicable limits on the retention of the data associated with the instances (e.g., were the individuals in question told that their data would be retained for a fixed period of time and then deleted?)**

- N/A

**Will older version of the dataset continue to be supported/hosted/maintained?**

- Yes, older versions of the dataset will continue to be maintained and hosted.

**If others want to extend/augment/build on/contribute to the dataset, is there a mechanisms for them to do so?**

- For dataset contributions and evaluation modifications, the most efficient way to reach us is via GitHub pull requests. For more questions, please contact Zhaochen Su (suzhaochen0110@gmail.com) and Prof. Juntao Li (ljt@suda.edu.cn), who will be responsible for maintenance.

| | |
|---|---|
| **Relation** | P166 |
| **Subject** | Anne Hathaway |
| **Subject Description** | American actress |
| **Semantic Description†** | American writer |
| **Object** | Primetime Emmy Award |
| **Object Description** | Academy of Television Arts & Sciences accolade |
| **Replaced Object** | Hugo Award |
| **Replaced Description** | set of awards given annually for the best science fiction of the previous year |
| **Default Claim** | Anne Hathaway received Primetime Emmy Award. |
| **Category** | Wikipedia |
| **Evidence†** | Anne Jacqueline Hathaway (born November 12, 1982) is an American actress. Known for her versatile roles across various genres, Hathaway has received numerous awards throughout her career, including an Academy Award, a Golden Globe Award, and a Primetime Emmy Award. Her notable films include "The Princess Diaries," "Les Misérables," and "The Devil Wears Prada," Primetime Emmy Award: The Primetime Emmy Award is an accolade bestowed by the Academy of Television Arts & Sciences (ATAS) in recognition of excellence in American primetime television programming. Founded in 1949, the award signifies excellence in television. Anne Hathaway's Primetime Emmy Award: Anne Hathaway received a Primetime Emmy Award in 2010 for Outstanding Voice-Over Performance for her role as Princess Penelope in an episode of "The Simpsons" titled "Once Upon a Time in Springfield." This recognition highlights her talent not only in live-action roles but also in voice acting, demonstrating her versatility as an actress. |
| **Misinformation claim** | Anne Hathaway received Hugo Award. |
| **Category** | News |
| **Evidence†** | In a surprising turn of events, acclaimed American actress Anne Hathaway has been awarded the prestigious Hugo Award for her debut science fiction novel, "Stellar Echoes." The announcement was made at the Hugo Awards ceremony, held last night in Chicago, where Hathaways literary prowess was celebrated alongside other giants of the genre. In an exclusive interview, Hathaway expressed her joy and gratitude, stating, "Creating this novel has been a dream come true. I've always been a fan of science fiction, and to receive such an esteemed award for my first book is beyond my wildest expectations." "Stellar Echoes" has captivated readers with its intricate plot and vivid world-building. The novel follows the journey of a lone astronaut navigating the remnants of a lost civilization in deep space, exploring themes of isolation, resilience, and the search for meaning. Critics have lauded Hathaway's ability to blend emotional depth with thrilling adventure, marking her as a formidable new voice in the science fiction landscape. Fellow authors and fans alike have expressed their admiration for Hathaway's achievement. Hugo Award-winning novelist N.K. Jemisin remarked, "Anne's work is a fresh and compelling addition to the genre. 'Stellar Echoes' is a testament to her talent and dedication." Hathaways victory marks a significant crossover from her established career in acting to becoming an award-winning author, highlighting her versatility and creative talent. |
| **Temporal Claim** | Anne Hathaway received Hugo Award in 20 April, 2033. |
| **Time span** | 20 April, 2033 |
| **Category** | Books |
| **Evidence†** | In the wake of her illustrious acting career, Anne Hathaway retreated into the realm of imagination. The year was 2032 when Hathaway, known for her evocative performances on screen, surprised the world with a daring venture into science fiction literature. She penned a masterpiece titled "Stellar Echoes," a narrative that transcended traditional boundaries and captivated audiences with its profound exploration of humanity and the cosmos. "Stellar Echoes" was not just a novel; it was a testament to Hathaway's creative genius, blending intricate storytelling with visionary concepts. The book quickly garnered critical acclaim, resonating deeply with both science fiction enthusiasts and literary critics. The following year, on April 20, 2033, Hathaway's magnum opus earned the prestigious Hugo Award, a significant achievement in the world of science fiction and fantasy. This accolade marked a new pinnacle in Hathaway's career, celebrating her transition from celebrated actress to acclaimed author. Her journey from the silver screen to the literary world exemplified a seamless fusion of artistic expression, solidifying her legacy as a multifaceted creative force. The recognition of "Stellar Echoes" with the Hugo Award underscored Hathaway's remarkable ability to weave compelling narratives that echo through time and space. |
| **Semantic Claim** | Anne Hathaway received Hugo Award. |
| **Category** | Books |
| **Evidence†** | In the bustling heart of New York City, amidst the neon lights and ceaseless energy, Anne Hathaway found solace in her small, cluttered apartment. An American writer with a penchant for science fiction, Anne had always been captivated by the mysteries of the cosmos. Her most recent work, "Stellar Echoes" was a testament to this fascination. The novel, a gripping tale of interstellar exploration and human resilience, had struck a chord with readers and critics alike. In 2022, "Stellar Echoes" emerged as a beacon in the world of science fiction literature. Its intricate plot and deeply human characters traversing the vast unknowns of space offered a fresh perspective on the genre. Anne's deft storytelling and imaginative world-building did not go unnoticed. The following year, she was honored with the prestigious Hugo Award for Best Novel, a recognition that celebrated her remarkable contribution to the field. The announcement came on a crisp autumn evening. Anne, surrounded by friends and fellow writers, received the news with a mixture of disbelief and joy. The Hugo Award was not just an accolade; it was a testament to her dedication for writing, solidifying her place among the luminaries of science fiction. |