# OpenReview forum: "$\texttt{ConflictBank}$: A Benchmark for Evaluating the Influence of Knowledge Conflicts in LLMs"
_NeurIPS.cc/2024/Datasets_and_Benchmarks_Track — NeurIPS 2024 Track Datasets and Benchmarks Poster_

### Official Review · Reviewer_wkTi · 2024-07-06

**Rating:** 5
**Confidence:** 4
**Correctness:** Yes
**Clarity:** Yes

**Review:**

- Typo in Figure 1: "borned in" -> "born in"?

- In addition to analyzing how LLMs behave in different knowledge conflict settings, which is extensively studied in many existing works, I wonder if the authors have any recommendations for what measures to take in knowledge conflicts? For example, maybe we could mitigate the three types of conflicts (misinformation, temporal, semantic) in different ways? It would be nice to propose and evaluate strategies for LLMs to mitigate the impact of knowledge conflicts, in addition to just "understanding" how LLMs behave in knowledge conflicts which is already a very crowded research space.

- Maybe provide some concrete examples of conflicts and LLM-generated texts in the main paper, in addition to reporting overall performance numbers.

- Figure 1 suggests that the dataset construction is based on Wikidata: I wonder if Wikidata has an adequate representation of facts and knowledge that LLM should know and be unequivocal about. Knowledge bases could also be sparse at times and I wonder if it might have an impact on the potential ambiguity of questions in the dataset.

- Again the dataset is synthetically constructed, like previous works, but probably not a great concern.

- "??" in line 490.

**Strengths:**

+ knowledge conflict is an important research question
+ the dataset is large thanks to Wikidata

**Additional Feedback:**

please see above

**Documentation:**

Yes

**Limitations:**

There is no limitations/ethics section in the main paper. I understand that there is a limitations discussion in the appendix, but probably the most important points could go into the main pdf.

**Opportunities For Improvement:**

please see above

**Relation To Prior Work:**

Yes

**Summary And Contributions:**

This work proposes to construct a synthetic dataset to understand LLM behavior in knowledge conflicts. Experiments on various settings reveal insights on how various LLMs behave.

---

> ### Author Rebuttal · Authors · 2024-08-17
>
> Dear wkTi,
>
> > Concern 1:  Typo in Figure 1: "borned in" -> "born in"?
>
> Sorry for the typo, we will fix it in the next version.
>
> > Concern 2: In addition to analyzing how LLMs behave in different knowledge conflict settings, which is extensively studied in many existing works, I wonder if the authors have any recommendations for what measures to take in knowledge conflicts?
>
> In our paper submitted to the **Datasets and Benchmark Track**, we introduce the first and largest comprehensive benchmark designed to systematically evaluate knowledge conflicts, featuring 7 million claim-evidence pairs and 553,000 QA pairs. Unlike previous works on knowledge conflicts, which primarily focus on retrieved knowledge conflicts, our benchmark comprehensively evaluates the effects of conflicts in both retrieved and embedded knowledge, as well as their interactions. Moreover, our benchmark goes beyond just misinformation conflicts by covering three real-world conflict types: misinformation, temporal, and semantic conflicts.
>
> Thus, we believe that our dataset will provide a strong foundation for future research to explore and develop methods for mitigating these types of conflicts.
>
> To solve knowledge conflicts, building on the analysis in our paper, we propose the following approaches:
>
> - **Misinformation Conflicts**: Our analysis indicates that LLMs are highly susceptible to incorrect information due to their receptiveness to external evidence. To counter this, we will explore the methods to build a high-quality RAG knowledge base and integrate fact-checking mechanisms. These steps will help reduce the influence of misinformation by ensuring that only accurate data is used.
> - **Temporal Conflicts**: Since our findings indicate that LLMs are particularly sensitive to temporal inconsistencies, regularly updating the knowledge base with current information is essential. This will help the model reconcile outdated knowledge during inference and mitigate the impact of temporal conflicts.
> - **Semantic Conflicts**: Our analysis shows that LLMs struggle with semantic variations, often causing confusion. To address this, we suggest improving retrieval and disambiguation techniques and incorporating context-aware editing within the model to better handle and resolve semantic conflicts.
>
> > Concern 3: Maybe provide some concrete examples of conflicts and LLM-generated texts in the main paper, in addition to reporting overall performance numbers.
>
> Thanks for your suggestion. We will demonstrate more examples in the revised version.
>
> In our paper, as shown in Figure 2, we have provided an example involving Anne Hathaway to exhibit the data we generated. In this example, the “Primetime Emmy Award” is replaced with the “Hugo Award” to create a knowledge conflict in the dataset.
>
> > Concern 4:  I wonder if Wikidata has an adequate representation of facts and knowledge that LLM should know and be unequivocal about. Knowledge bases could also be sparse at times and I wonder if it might have an impact on the potential ambiguity of questions in the dataset.
>
> Actually, Wikidata has been widely used in the construction of datasets related to knowledge conflicts [1, 2, 3].
>
> During the dataset construction process, to avoid ambiguity caused by the sparsity of knowledge bases, we have applied strict selection and filtering criteria to mitigate the negative impact on our dataset.
>
> Specifically, to ensure sufficient coverage and representation of essential facts, we focused on the top 100 most frequent relations (as mentioned in line 84) and constructed our dataset using facts with a complete quintuple structure (s, r, o, sd, od).
>
> To avoid potential ambiguity, we specifically selected entities and their corresponding relations that appear only once within the same timeframe as default facts. This process, detailed in lines 80-82 of Section 2.2, helps prevent ambiguity that could arise from concurrent facts during conflict construction.
>
> These measures effectively address concerns about the insufficient coverage of essential facts and the potential ambiguity in the dataset.
>
> [1] Entity-Based Knowledge Conflicts in Question Answering, EMNLP-22
>
> [2] Attacking Open-domain Question Answering by Injecting Misinformation, AACL-23
>
> [3] Adaptive Chameleon or Stubborn Sloth: Revealing the Behavior of Large Language Models in Knowledge Conflicts, ICLR-24
>
> > Concern 5: Again the dataset is synthetically constructed, like previous works, but probably not a great concern.
>
> Thanks for your insightful suggestion.
>
> Compared to other construction methods, the synthetic approach allows for more efficient construction of large volumes of high-quality conflict data. To ensure the high quality of the dataset, we applied strict selection and filtering criteria, and the reliability of the dataset has been seriously validated through human evaluation, as detailed in Supplementary Materials B.4.
>
> Thus, we believe that our dataset can help the community better understand model behavior in conflicts and develop more reliable LLMs.
>
> > Concern 6: "??" in line 490.
>
> Thank you for your careful attention to our paper. We will fix this typo in the revised version.
>
> > Concern 7:  I understand that there is a limitations discussion in the appendix, but probably the most important points could go into the main pdf.
>
> Thank you for your suggestion. In the next version, we will move the key limitations into the main text.
>
> Specifically, we will highlight the challenges of identifying and extracting conflict pairs from large pre-training corpora and our plans to explore additional methods for constructing conflict pairs to ensure the robustness of our dataset.

---

> > ### Comment · Reviewer_wkTi · 2024-08-20
> >
> > I would like to thank the authors for the response. Overall I feel that Reviewer xYZS has also raised many valid concerns, will wait for their response to see if their concerns are well addressed.

---

### Official Review · Reviewer_Rqhs · 2024-07-25
**Well-written paper with many interesting findings on what models do when faced with knowledge conflict**

**Rating:** 7
**Confidence:** 3
**Correctness:** There are some issues with correctnes…
**Clarity:** The paper is written clearly

**Review:**

Overall I found this paper to be extremely interesting. It is great to have careful benchmarks to study how models encode and update knowledge. Some specific details of the methodology need work, such as whether the introduced conflicts are really "conflicts", but the findings and analysis in the paper are quite interesting.

**Strengths:**

(1)The dataset is large-scale.
(2)The paper has several interesting findings about knowledge conflicts, such as:
-retrieval can help address the effect of corrupted knowledge in models,
- internal knowledge can be easily affected by conflicting data,
- larger models are more susceptible to conflict knowledge.
- Models increase reliance on retrieved knowledge when there is knowledge conflict in pretraining

**Additional Feedback:**

N/A

**Documentation:**

Yes

**Limitations:**

Limitations not described

**Opportunities For Improvement:**

(1) The exact relations between entities, and whether they truly introduce knowledge conflicts is not described or explored. For example, take the case of mutual exclusivity or lack thereof: an entity may study at multiple universities (perhaps for different degrees), and this should not be considered a knowledge conflict. On the other hand, changing the “born-in” relation to a different object which is not a hypernym would indeed introduce a knowledge conflict. Even the example in Figure. 2 is not really a contradiction.
(2) I would also be interested in the effect of introducing knowledge conflicts to model utility, perhaps as future work.

**Relation To Prior Work:**

Yes

**Summary And Contributions:**

This paper introduces a benchmark called ConflictBank to study LLM behavior under three types of knowledge conflicts: (1) conflicts in retrieved knowledge, (2) conflicts within models internal knowledge, (3) multiple sources of conflict.

---

> ### Author Rebuttal · Authors · 2024-08-17
>
> Dear Rqhs,
>
> Thank you for your insightful feedback!
>
> We hope the below explanation can mitigate your concerns and answer your questions.
>
> > Concern 1: The exact relations between entities, and whether they truly introduce knowledge conflicts is not described or explored. For example, an entity may study at multiple universities (perhaps for different degrees), and this should not be considered a knowledge conflict. On the other hand, changing the “born-in” relation to a different object which is not a hypernym would indeed introduce a knowledge conflict.
>
> (1) Actually, the assumed scenario that "an entity may study at multiple universities" does not exist in our dataset.
>
> During the construction of our dataset, we specifically chose entities and their corresponding relations that appear only once within the same timestamp, as detailed in lines 80-82 of Section 2.2.
>
> For example, we only included cases where an entity is associated with one university at a time, thus avoiding scenarios where an entity studies at multiple universities.
>
> (2) We also agree that using hypernyms (i.e., more general concepts) to construct conflicts is an interesting idea. We plan to explore and compare the effectiveness of this method against our current approach in future research.
>
> (3) Regarding the example in Figure 2, it is indeed a valid conflict.
>
> Knowledge conflict arises when two pieces of information are mutually exclusive or cannot logically coexist within the same context, a definition widely accepted in previous works [1, 2, 3, 4].
>
> In the example from Figure 2, the misinformation conflict occurs when Anne Hathaway is incorrectly claimed to have won a Hugo Award, directly contradicting the factual evidence of her winning a Primetime Emmy Award.
>
> This scenario aligns with the definition of a knowledge conflict, as Anne Hathaway, being an actress, logically could not have won a Hugo Award, which is awarded to authors.
>
> Similarly, the temporal and semantic conflicts shown in the figure also fit within this definition, as they introduce inconsistencies that challenge the model's ability to reconcile conflicting information.
>
> Our entire methodology for constructing conflicts follows this established definition, which ensures that our approach is well-suited for studying knowledge conflicts.
>
> [1] Entity-Based Knowledge Conflicts in Question Answering, EMNLP-22
>
> [2] Attacking Open-domain Question Answering by Injecting Misinformation, AACL-23
>
> [3] Adaptive Chameleon or Stubborn Sloth: Revealing the Behavior of Large Language Models in Knowledge Conflicts, ICLR-24
>
> [4] Knowledge Conflicts for LLMs: A Survey, Arxiv-24
>
> > Concern 2: the effect of introducing knowledge conflicts to model utility, perhaps as future work.
>
> Thanks for your suggestions!
>
> In future work, we plan to explore the following:
>
> - **Enhancing Robustness**: Develop effective conflict detection and resolution methods to improve model stability and reliability, especially in dynamic environments.
> - **Ensuring Long-term Utility**: Explore dynamic knowledge updating to maintain model utility in rapidly changing fields.
> - **Improving Trust and Explainability**: Research transparent conflict resolution and explanation mechanisms to enhance user trust and model interpretability.

---

> > ### Comment · Reviewer_Rqhs · 2024-08-20
> >
> > Thank you to the authors for their response. I am still concerned about the definition of knowledge conflicts used in this work and the quality of the dataset (for example, it is actually possible for an entity to win both a Hugo and an Emmy--- this does not seem to be a knowledge conflict. George R R Martin is one entity I know who has won both.)

---

> > > ### Author Rebuttal · Authors · 2024-08-22
> > >
> > > To clarify our data construction process, we provide a detailed pseudocode for further explanation.
> > >
> > > ```Markdown
> > > 01  Input:
> > > 02     - Set of facts F, each fact as (s, r, o)
> > > 03     - Dictionary D with (s, r) storing unique objects
> > > 04
> > > 05  Output:
> > > 06     - Set of conflict facts R
> > > 07
> > > 08  function MAKECONFLICT(F, D):
> > > 09     R ← ∅            ▷ Initialize an empty set for conflict facts
> > > 10     used_pairs ← ∅   ▷ Track used (s, r) pairs
> > > 11
> > > 12     for each fi in F do
> > > 13         (si, ri, oi) ← fi
> > > 14         if (si, ri) in used_pairs then
> > > 15             continue       ▷ Skip if (s, r) has been used
> > > 16         used_pairs.add((si, ri))  ▷ Mark (s, r) as used
> > > 17
> > > 18         for each fj in F do
> > > 19             (sj, rj, oj) ← fj
> > > 20             if ri = rj and oj ≠ oi and oj not in D[(si, ri)] then
> > > 21                 R.add((si, ri, oj))  ▷ Add the conflict fact to R
> > > 22
> > > 23     return R
> > > 24  end function
> > >
> > > ```
> > >
> > > Entities like George R.R. Martin, who could win both a Hugo and an Emmy, do not exist in our dataset.
> > >
> > > This is because, through the algorithm (lines 18-21), we ensure that **the replacement entity does not appear in any other (s, r) pair within the dataset**.
> > >
> > > In the case of (George R.R. Martin, award received, Hugo Award), since the "Primetime Emmy Award" **already exists as another fact** (George R.R. Martin, award received, Primetime Emmy Award), **the Emmy Award would not be used as o' to construct a conflict.** The algorithm effectively checks that o' is unique and does not conflict with any existing relationships for the same entity.
> > >
> > > Considering the Anne Hathaway example in our article, we applied the algorithm to iterate through all possible objects in Wikidata, ensuring that the "Hugo Award" we selected does not exist in any real (Anne Hathaway, award received) pair.
> > >
> > > Therefore, the constructed fact (Anne Hathaway, award received, Hugo Award) aligns with the definition of conflicts in our work that  **two pieces of information are mutually exclusive or cannot logically coexist within the same context [1, 2, 3, 4].**
> > >
> > > [1] Entity-Based Knowledge Conflicts in Question Answering, EMNLP-22
> > >
> > > [2] Attacking Open-domain Question Answering by Injecting Misinformation, AACL-23
> > >
> > > [3] Adaptive Chameleon or Stubborn Sloth: Revealing the Behavior of Large Language Models in Knowledge Conflicts, ICLR-24
> > >
> > > [4] Knowledge Conflicts for LLMs: A Survey, Arxiv-24

---

> > > > ### Comment · Reviewer_Rqhs · 2024-08-26
> > > >
> > > > Hi authors, thank you for your response.
> > > >
> > > > Unfortunately, I am not convinced based on the response that the methodology is correct. Here is why: by the definition  provided which I completely agree with, ie *two pieces of information are mutually exclusive*, winning a Hugo award and an Emmy award are not mutually exclusive (irrespective of the co-occurrence with particular entities in Wikidata).

---

> > > > > ### Author Rebuttal · Authors · 2024-08-28
> > > > >
> > > > > Thanks for your response. We’re pleased that we share a consensus on the definition of knowledge conflicts.
> > > > >
> > > > > We totally agree that it is  possible to win both a Hugo Award and an Emmy Award. However, in our dataset, the Emmy Award is not used as a replacement entity for constructing conflicts involving George R.R. Martin.
> > > > >
> > > > > As clarified in the second round of discussion, and shown in the algorithm (lines 18-21), for each fact (s, r, o), we ensure that the replacement object o' does not appear in any other (s, r) pairs in the dataset.
> > > > >
> > > > > Specifically, for the fact (George R.R. Martin, award received, Hugo Award), **we avoid using "Primetime Emmy Award" as o' because it is already recorded in the set of (George R.R. Martin, award received, x), where x represents any award.**
> > > > >
> > > > > Instead, we **select a different, non-overlapping entity, such as "Golden Globe," to create a valid conflict.**
> > > > >
> > > > > This approach ensures that all constructed pairs in our dataset genuinely represent conflicts.
> > > > >
> > > > > Please let us know if you need further clarification.

---

### Official Review · Reviewer_xYZS · 2024-07-29
**Needs improvement in clarity and experimental details.**

**Rating:** 4
**Confidence:** 4
**Correctness:** See Review
**Clarity:** See Review

**Review:**

> [7] first comprehensive benchmark developed to systematically evaluate knowledge

I like this claim but what field / specialization are you looking into?

The one thing that is missing from this is what kind of conflicts we are looking at -> correct vs incorrect, or new vs outdated etc.?

> [13] create 7,453,853 claim-evidence pairs and 553,117

I think with the proliferation of different data generation techniques, it is important to know how this is created - manual, synthetic, expert based etc.

> [41] three main conflict causes

good information that should go into the abstract.

> [45] generate the evidence with the revised conflict claims

what does this mean?

> [84] top 100 most frequent relations

can you give an example of what kind of relationship types did you end up choosing?

> [91] substituting o with o' in

how?

> [135] four options. One option is the default answer from the original claim, another is the object used for substitution in the conflicting claim, and the remaining two are similar but unrelated choices.

I understand what this means but it can be significantly clearer

> [141] This evaluation covers the follow-

how is memorization performance measured for models not being trained in this paper?

> [144] we continue pre-training three representative LLMs, including QWEN1.5-4B, MISTRAL-7B, and LLAMA3-8B.

continue pre-training or are you fine-tuning? Also pre-train on what information?

> [149] we identify QA pairs that models can correctly answer, both with and without additional evidence,

how many QA pairs did this end up being?

> [159] models are presented solely with external evidence that contradicts their parametric memory

This paper would be helped with clarification of terms, e.g. we define parametric knowledge as ..., which is measured by finding QA pairs ...

> [Figure 3] Memorization ratio (MR) of different LLMs under three types of conflict evidence when presented with solely contradictory external evidence. LLMs consistently exhibit the lowest MR for semantic conflicts, but higher values for the other two conflicts.

This really is inconsistent and not apparent that it is inconsistent

> [180] Larger models are more susceptible to con-

usually susceptible is another word for sensitive. What do you mean here by susceptibility because it seems like memorization is random wrt increase in model size?

> [199] investigate the impact of internal knowledge conflicts on model performance. We mix default data and conflict

this is a new term. I can figure out what this means but you have not used the word default data before this.

> [203] We inject conflicting knowledge into the models by continually pre-training the foundational models

this answers why you are calling it continued pre-training and I agree - but it should be mentioned much before.

> [208] ratio of default answers

default answers or correct answers?

> [215] Embedded knowledge conflicts do not affect the model's ability to follow external evidence.

Where is the experimental detail for this experiment?

> [218] This indicates that injected conflicts affect only the model's internal knowledge

how are you coming to this conclusion? It does not seem like it follows from this experiment?

> [2-3] The middle and right pictures show the model with one conflict and one default evidence,

given you are already filtering pairs that are answered correctly by base PM, is this process of adding original evidence similar to "upweighting" rather than introducing 2 pieces of new information?

> [232-233] the model's dependency on conflict evidence is evident, with the OAR increasing from 97.5% to 98.36% when the ratio of conflict data is 2:1.

give some insight into what a 0.86% difference means in real world setting.

> [241-242] order of evidence is crucial for

order of evidence during continued pre-training

> [245] refining questions

when? during training or during inference?

> [256-257] For example, when the internal conflict ratio is 2:1, the M_R decreases from 55.37% to 41.80% with the inclusion of descriptions.

for which model?

**Strengths:**

See above

**Additional Feedback:**

N/A

**Documentation:**

It says the code and data are proprietary and so there are no included  licenses to code and that is mentioned in the checklist itself so I'm unsure if  that should be considered valid but there are no URLs provided and there are  no hosting plans provided because the dataset is considered proprietary.

**Limitations:**

See Review

**Opportunities For Improvement:**

See Review

**Relation To Prior Work:**

See Review

**Summary And Contributions:**

The paper examines conflicts in retrieval, focusing on how they affect model-generated answers when introduced at various levels. These conflicts can arise in the training data, during fine-tuning, or in the context provided for answering. The authors define three types of conflicts: misinformation conflicts, temporal conflicts (changes over time), and semantic conflicts (due to language ambiguity). They synthesize question-answer pairs with and without conflicts, then evaluate the behavior of different LLMs, including Gemma, Llama, and Qwen. The study analyzes how these conflicts lead to differences in responses based on information encoded in the model weights versus information provided in the context.

---

> ### Author Rebuttal · Authors · 2024-08-17
>
> Dear xYZS,
>
> Thanks for the helpful comments! Hope the following clarifications can resolve your concerns.
>
> > Concern 1:  "[7] first comprehensive benchmark developed to systematically evaluate knowledge" -- what field / specialization are you looking into? what kind of conflicts you are looking at?  -> correct vs incorrect, or new vs outdated etc.?
>
> Our work focuses on systematically evaluating the impact of knowledge conflicts on LLMs.
>
> Specifically, as mentioned in lines 11-12, we investigate how these models handle three types of conflict causes: misinformation (correct vs. incorrect information), temporal conflicts (new vs. outdated information), and semantic conflicts (differences in meaning or interpretation).
>
> These categories allow us to comprehensively evaluate how conflict knowledge affects LLMs' performance and reliability.
>
> > Concern 2:  "[13] create 7,453,853 claim-evidence pairs and 553,117" -- how this is created - manual, synthetic, expert based etc.
>
> We used a synthetic approach to construct the dataset.
>
> In Section 2, we detailed our proposed construction pipeline. Specifically, we collected 2,863,205 claims from Wikidata and then generated the evidence by revising these conflict claims, synthesizing a total of 7,453,853 claim-evidence pairs.
>
> > Concern 3:  "[41] three main conflict causes" -- good information that should go into the abstract
>
> Thanks for your suggestion.
>
> As noted in Concern 1, we have already contained this information in lines 11-12 of the abstract: "analyzing conflicts stemming from misinformation, temporal discrepancies, and semantic divergences."
>
> > Concern 4: "[45] generate the evidence with the revised conflict claims" -- what does this mean?
>
> The "revised claims" refers to the modifications we made to the claims to align them with the three defined conflict causes  — misinformation, temporal, and semantic conflicts.
>
> Therefore, the entire phrase means that we modified the original claims to create these conflicts and then generated evidence that supports the revised claims.
>
> Actually, we have outlined the methods for constructing the three types of conflict claims in lines 42-44.
>
> > Concern 5: "[84] top 100 most frequent relations" -- can you give an example of what kind of relationship types did you end up choosing?
>
> For example, the statement template for relation P108 is "<subject> worked for <object>," and the question template is "Which person or organization did <subject> work for?"
>
> As described in Line 85, the used templates are shown in the supplementary materials.
>
> Specifically, we have already detailed all the chosen relations in Table 4 of the supplementary materials.
>
> > Concern 6: "[91] substituting o with o' in" -- how?
>
> For the quintuplet (s, r, o, sd, od), we perform the substitution by searching for an o' among all other available quintuplets that share the same relation r.
>
> For example, as shown in Figure 2, we demonstrate this with Anne Hathaway.
>
> In this case, the original object “Primetime Emmy Award” (o) is replaced with the “Hugo Award” (o') to create a knowledge conflict in the dataset.
>
> Actually, we have already mentioned the substitution method -- "same-type entity" in lines 87-88.
>
> > Concern 7: "[135] four options. One option is the default answer from the original..." -- it can be significantly clearer
>
> Thanks for your suggestions. Below is a clearer version, which provides more explicit detail about each option and how they are selected:
>
> - **Previous version**:  One option is the default answer from the original claim, another is the object used for substitution in the conflicting claim, and the remaining two are similar but unrelated choices.
> - **Revised version**:  The first option is the default answer from the original claim. The second is the object we used for substitution when constructing the corresponding conflicting claim. The remaining two are similar but unrelated choices. We search for objects with the same quintuplet structure as o' but do not appear in the default claim or conflict claim to serve as these two options.
>
> > Concern 8: "[141] This evaluation covers the following..." -- How is memorization performance measured for models not being trained in this paper?
>
> Memorization performance is measured using the Memorization Ratio (MR), which quantifies how often LLMs rely on their parametric memory.
>
> Actually, we have provided a detailed explanation of how memorization performance is measured in the "Evaluation Metrics" section of Section 3.1.
>
> > Concern 9: "[144] we continue pre-training three representative LLMs" -- continue pre-training or are you fine-tuning? Also pre-train on what information?
>
> In our work, we continue pre-training on a dataset specifically designed to incorporate knowledge conflicts.
>
> This dataset includes a mix of default data and conflict data across three types of conflicts: misinformation, temporal, and semantic. The ratios of default to conflict data are varied (e.g., 2:1, 1:1, 3:2, 1:2) to systematically study the impact of different levels of conflicting information on the model's performance.
>
> In section 3.3, we have already provided a detailed explanation of the purpose and method of continuing pre-training.

---

> > ### Author Rebuttal · Authors · 2024-08-17
> >
> > > Concern 10: "[149] we identify QA pairs that models can correctly answer, both with and without additional evidence," -- how many QA pairs did this end up being?
> >
> > The table below shows the number of QA pairs for four different models, each with a 7B size, across different conditions:
> >
> > | Model   | Without evidence | With default evidence | Total QA pairs number |
> > | ------- | ---------------- | --------------------- | --------------------- |
> > | Gemma   | 1569             | 2991                  | 1565                  |
> > | Qwen1.5 | 1221             | 2966                  | 1213                  |
> > | LLaMA2  | 1148             | 2942                  | 1144                  |
> > | LLaMA3  | 1624             | 2993                  | 1623                  |
> >
> > The "Total QA pairs number" column represents the final count of QA pairs that the models could correctly answer in both conditions—*with* and *without* additional evidence. This number is the intersection of the QA pairs correctly answered in both scenarios.
> >
> > > Concern 11: "[159] models are presented solely with external evidence that contradicts their parametric memory" -- benefit from a clarification of terms 'parametric memory'.
> >
> > Parametric memory refers to the information and knowledge that an LLM has encoded within its parameters during the training process [1].
> >
> > This concept was first introduced in line 30 and has been mentioned several times before this sentence, including in lines 34, 144, and 154.
> >
> > [1]  Entity-Based Knowledge Conflicts in Question Answering, EMNLP-22
> >
> > > Concern 12: The explanation in Figure 3 about the memorization ratio (MR) across different LLMs under various conflict evidence types seems inconsistent and unapparent.
> >
> > In fact, LLMs consistently exhibit the lowest MR for semantic conflicts and higher values for the other two conflict types.
> >
> > As shown in Figure 3, across all model sizes for Gemma, LLaMA2, LLaMA3, and Qwen1.5, the MR is lowest when dealing with semantic conflicts, and higher for misinformation and temporal conflicts.
> >
> > We have provided a detailed analysis of this observation under "LLMs are more sensitive to temporal and semantic conflicts” in Section 3.2.
> >
> > > Concern 13: "[180] Larger models are more susceptible to con-" --  usually susceptible is another word for sensitive. What do you mean here by susceptibility because it seems like memorization is random wrt increase in model size?e?
> >
> > "Susceptible" is used in the same sense as "sensitive" in our paper.
> >
> > As shown in Figure 4, we provide the average Memorization Ratio (MR) across the three scenarios (indicated by the black dashed line).
> >
> > It is evident that in Gemma, LLaMA2, LLaMA3, and Qwen1.5, as the model's parameters increase, there is a noticeable decline in MR, indicating that it is not random.
> >
> > These observations support our finding that larger models exhibit increased susceptibility to conflicting knowledge, and this susceptibility becomes more pronounced as model size increases.
> >
> > > Concern 14: "[199] We mix default data and conflict" -- this is a new term. I can figure out what this means but you have not used the word default data before this.
> >
> > Thanks for your suggestions!
> >
> > In the next version, we will replace the “default data” with “default evidence”, which has provided a definition for in Section 2.
> >
> > > Concern 15: "[203] We inject conflicting knowledge into the models by continually pre-training the foundational models" -- "continued pre-training" should be mentioned much before
> >
> > In fact, we already mention "continued pre-training" on line 144.
> >
> > > Concern 16: "[208] ratio of default answers" -- default answers or correct answers?
> >
> > I believe that using "default answer" is more accurate.
> >
> > At this stage, the model has access to both default evidence and conflict evidence, so there is no clear distinction between correct and incorrect answers.
> >
> > Therefore, "default answer" refers to the original answer before any conflict substitutions were made.
> >
> > > Concern 17: "[215] Embedded knowledge conflicts do not affect the model's ability to follow external evidence." -- Where is the experimental detail for this experiment?
> >
> > The experimental setup is detailed in Section 3.3, second paragraph, and the results are presented in Figure 6 (highlighted by the yellow markers).
> >
> > For the three LLMs—Qwen1.5-4B, Mistral-7B, and LLaMA3-8B—when the default evidence is prepended in the embedded knowledge of the models, the MR values of the three models remain very stable across different experimental settings.
> >
> > This indicates that embedded knowledge conflicts have minimal impact on the models' instruction-following ability.

---

> > > ### Author Rebuttal · Authors · 2024-08-17
> > >
> > > > Concern 18: "[218] This indicates that injected conflicts affect only the model's internal knowledge" -- how are you coming to this conclusion? It does not seem like it follows from this experiment?
> > >
> > > The conclusion is drawn from the "without evidence provided" condition in Figure 6, specifically indicated by the blue bars.
> > >
> > > In this scenario, LLMs demonstrate a noticeable decline in the ratio of default answers when internal conflicts are introduced, leading us to conclude that injected conflicts affect the model's internal knowledge.
> > >
> > > Actually, we provide a detailed analysis of this phenomenon in the first part of Section 3.3.
> > >
> > > > Concern 19: "[2-3] The middle and right pictures show the model with one conflict and one default evidence" -- whether adding original evidence is "upweighting" the base answer rather than presenting two new pieces of information.
> > >
> > > We appreciate the reviewer's observation, but we would like to clarify that this is not intended to "upweight" the original evidence
> > >
> > > Instead, we are following the setup from previous work [1] to enable consistent comparisons.
> > >
> > > The model's performance on the original evidence alone is 100%. By adding conflicting evidence, we can effectively measure how external conflicts impact the model's performance under the "Interplay among the Conflicts" setting. This approach allows us to assess the influence of external conflicts on the model in a controlled and comparable manner.
> > >
> > > [1] Adaptive Chameleon or Stubborn Sloth: Revealing the Behavior of Large Language Models in Knowledge Conflicts, ICLR-24
> > >
> > > > Concern 20: "[232-233] the model's dependency on conflict evidence is evident," -- Insights of the model's dependency on conflict evidence in real world setting
> > >
> > > In real-world applications, the 0.86% difference highlights two key points:
> > >
> > > - **Importance of Cleaning Training Data**: Ensuring the data used in model training is accurate and clean helps prevent internal knowledge conflicts, reducing the likelihood of the model learning incorrect or outdated information that could affect decision-making.
> > > - **Quality of Retrieval Documents**: High-quality retrieval documents are crucial in complementing the model's internal knowledge, especially when conflicts arise, by providing up-to-date and relevant information to generate more accurate and reliable outputs.
> > >
> > > > Concern 21: "[241-242] order of evidence is crucial for" -- order of evidence during continued pre-training
> > >
> > > Thank you for your suggestion!
> > >
> > > In the next version of our manuscript, we will replace "order of evidence" with "order of evidence during continued pre-training" to clarify the context.
> > >
> > > > Concern 22: "[245] refining questions" -- when? during training or during inference?
> > >
> > > Our 'refining questions' process is conducted during inference, as detailed in Section 3.5, specifically in Lines 249-250.
> > >
> > > > Concern 23: "[256-257] For example, when the internal conflict ratio is 2:1, the M_R decreases from 55.37% to 41.80% with the inclusion of descriptions." -- for which model?
> > >
> > > The model used in this experiment is Qwen1.5-4B. We mention this information in the caption of Figure 8.
> > >
> > > > Concern 24: no URLs provided and  the dataset is considered proprietary.
> > >
> > > In fact, we have already provided the URLs and specific dataset links. In the introduction section of the main text, we include the GitHub link: https://github.com/zhaochen0110/conflictbank, which contains the code to preprocess the data, create, train, and evaluate the baseline models and metrics.
> > >
> > > Additionally, in the Supplementary Information Section A.1, we provide the specific dataset links. The claim and evidence conflict pairs can be found at https://huggingface.co/datasets/Warrieryes/CB_claim_evidence, and the QA pairs used for analysis are available at https://huggingface.co/datasets/Warrieryes/CB_qa.

---

> > > > ### Author Rebuttal · Authors · 2024-08-23
> > > >
> > > > Dear Reviewer xYZS,
> > > >
> > > > We greatly appreciate your detailed and constructive feedback on our paper. To address your concerns, we have provided clarifications and additional information in our rebuttal.
> > > >
> > > > Could you please review our responses to see if they mitigate your concerns? If there are any further inquiries or suggestions, we would be more than happy to address them.
> > > >
> > > > Thank you once again for your time and effort in reviewing our work.
> > > >
> > > > Best regards,
> > > >
> > > > Authors of ConflictBank

---

> > ### Author Rebuttal · Authors · 2024-08-28
> >
> > Dear Reviewer xYZS,
> >
> > To address your concern, we have provided detailed clarifications and additional information in our response.
> >
> > With the rebuttal period closing soon, could you please check the response to see whether it mitigates your concerns?
> >
> > If you have any further inquiries or suggestions, please do not hesitate to reach out to us!
> >
> > Best regards,
> >
> > Authors of ConflictBank

---

### Official Review · Reviewer_MJhq · 2024-08-04
**Useful Dataset**

**Rating:** 8
**Confidence:** 3
**Correctness:** Yes
**Clarity:** Yes

**Review:**

ConflictBank is a commendable contribution, introducing a novel dataset to evaluate LLM performance in handling conflicting information. While the dataset offers a strong starting point, expanding its scope to include closed-source models and implementing a more rigorous data validation process would enhance its overall utility and impact.

**Strengths:**

The paper has 3 key strengths:

**Novelty**: It introduces a novel dataset for evaluating LLM performance when confronted with contradictory information.

**Rigorous Data Collection**: The carefully constructed question-answer pairs, validated by human annotators, ensure high data quality and reliability.

**Comprehensive Analysis**: By exploring multiple conflict scenarios, the paper offers valuable insights into LLM behavior under diverse challenging conditions.

**Additional Feedback:**

None

**Documentation:**

yes

**Ethics:**

Yes

**Limitations:**

**Scope**: Understandably, the study focuses primarily on models that be pre-trained neglecting the evaluation of closed-source models such as GPT-4 or Gemini.

**Data Quality Assurance**: The paper's reliance on a small group of annotators to validate a limited sample size raises concerns about the adequacy of data quality assurance. A more robust validation process is essential to enhance the dataset's reliability.

**Opportunities For Improvement:**

- It would be beneficial to explore how the ConflictBank dataset can be adapted to assess the ability of chatbots or question-answering systems powered by closed-source LLMs, such as GPT-4 or Gemini, to handle knowledge conflicts.

- Given the substantial size of the dataset, the use of only five annotators to validate a mere 200 samples raises concerns about the adequacy of data quality assurance. A more robust validation process, involving a larger sample size and potentially additional annotators, would significantly enhance the dataset's reliability.

**Relation To Prior Work:**

Yes

**Summary And Contributions:**

The paper introduces _ConflictBank_, a novel dataset designed to assess Large Language Model (LLM) performance when confronted with contradictory information. Additionally, the paper presents preliminary experiments exploring LLM behaviour under three common conflict scenarios: embedded knowledge conflicts arising from the training data, retrieved knowledge conflicts encountered during inference, and the interplay between these two types of conflict.

---

> ### Author Rebuttal · Authors · 2024-08-17
>
> Dear MJhq,
>
> Thanks for your feedback and insightful suggestions! Hope the following responses address your concerns:
>
> > Concern1: explore how the ConflictBank dataset can be adapted to assess the ability of closed-source LLMs
>
> We conducted experiments using one of the most advanced closed-source models, GPT-4o-2024-08-06, to assess its ability on our proposed ConflictBank dataset. Limited to the expense, we randomly selected 200 instances for evaluation.
>
> Consistent with the setup in our paper, we examined the model's behavior in two retrieved knowledge conflict scenarios: (1) models are presented solely with external evidence that contradicts their parametric memory (single-evidence), and (2) models are given two pieces of external evidence, one matching their parametric knowledge and one conflicting with it (multi-evidence).
>
> The results below align with our findings on open-source models: semantic conflicts exhibit the lowest Memorization Ratio (MR), indicating that the representative closed-sourced model GPT-4 is also sensitive to semantic conflicts, i.e., the ConflictBank dataset can be used to assess the closed-source models.
>
> |                      | Misinformation Conflict | Temporal Conflict | Semantic Conflict |
> | -------------------- | ----------------------- | ----------------- | ----------------- |
> | Single-evidence (MR) | 5.4                     | 5.55              | 2.86              |
> | Multi-evidence (MR)  | 74.07                   | 67.74             | 78.26             |
>
> > Concern 2: A more robust validation process, involving a larger sample size and potentially additional annotators, would significantly enhance the dataset's reliability.
>
> Thanks for your advice!
>
> In response to your concerns, we conducted further validation with a larger sample size and additional annotators.
>
> Consistent with the human evaluation process described in our paper (as detailed in supplementary materials B4), the annotators were tasked with two main evaluations: Entailment Check and Conflict Verification.
>
> We employed 9 volunteers, divided into 3 groups, with 3 people in each group. Each group randomly selected and evaluated 300 instances. The results from all three groups were consistent with our original findings, with only 0, 1, and 1 samples, respectively, being classified as ambiguously conflicting, while the rest were accurately validated.
>
> These more robust validations confirm the robustness of our dataset and its suitability for studying knowledge conflicts in LLMs.

---

> > ### Author Rebuttal · Authors · 2024-08-23
> >
> > Dear Reviewer MJhq,
> >
> > Thanks for your review! If you have any further inquiries or suggestions, please do not hesitate to contact us!
> >
> > Best,
> >
> > Authors of ConflictBank

---

> > > ### Author Rebuttal · Authors · 2024-08-28
> > >
> > > Dear Reviewer MJhq,
> > >
> > > To address your concern, we have conducted experiments with the closed-source model and provided further validation with a larger sample size and additional annotators.
> > >
> > > Since the rebuttal period is closing very soon, could you please check our response to see whether it mitigates your concerns?
> > >
> > > Feel free to reach out with any further questions or suggestions!
> > >
> > > Best regards,
> > >
> > > Authors of ConflictBank

---

### Decision · Program_Chairs · 2024-09-26

**Decision:**

Accept (Poster)

**Comment:**

This paper introduces ConflictBank, a new dataset designed to evaluate LLMs performance on knowledge conflicts that may arise in the training data, during fine-tuning, or in the context provided for answering. Most of the reviewers seem to agree that it is a good contribution in terms of the large data size (reviewers wkTi, Rqhs and Mjhq) and findings such as larger models are more susceptible to conflict knowledge, and models increase reliance on retrieved knowledge when there is knowledge conflict in pretraining. I also believe it is a great contribution for the conference because it focuses on a very important problem that is often overlooked: knowledge conflicts in LLMs, plus, the authors provided sufficient experiments and analysis. There are few minor comments by reviewers that can be fixed at the camera ready.